# Text Quality-Based Pruning for Efficient Training of Language Models

**Vasu Sharma\***
FAIR, Meta
VASU@META.COM

**Karthik Padthe\***
FAIR, Meta
KARTHIKPADTHE@META.COM

**Newsha Ardalani**
FAIR, Meta
NEW@META.COM

**Kushal Tirumala**
FAIR, Meta
KTIRUMALA@META.COM

**Russell Howes**
FAIR, Meta
RHOWES@META.COM

**Hu Xu**
FAIR, Meta
HUXU@META.COM

**Po-Yao Huang**
FAIR, Meta
BERNIEHUANG@META.COM

**Shang-Wen Li**
FAIR, Meta
SHANGWEL@META.COM

**Armen Aghajanyan**
FAIR, Meta
ARMENAG@META.COM

**Gargi Ghosh**
FAIR, Meta
GGHOSH@META.COM

**Luke Zettlemoyer**
FAIR, Meta
LSZ@META.COM

**Reviewed on OpenReview:**

**Editor:**

## Abstract

In recent times training Language Models (LMs) have relied on computationally heavy training over massive datasets which makes this training process extremely laborious. In this paper we propose a novel method for numerically evaluating text quality in large unlabelled NLP datasets in a model agnostic manner to assign the text instances a *"quality score"*. By proposing the text quality metric, the paper establishes a framework to identify and eliminate low-quality text instances, leading to improved training efficiency for LM models. Experimental results over multiple models and datasets demonstrate the efficacy of this approach, showcasing substantial gains in training effectiveness and highlighting the potential for resource-efficient LM training. For example, we observe an absolute accuracy improvement of 0.9% averaged over 14 downstream evaluation tasks for multiple LM models

while using 40% lesser data and training 42% faster when training on the OpenWebText dataset and 0.8% average absolute accuracy improvement while using 20% lesser data and training 21% faster on the Wikipedia dataset.

**Keywords:** Natural Language Processing, Data pruning, Text Quality, Efficient Deep Learning, Large Language Models

# 1 Introduction

Language Models (LMs) have gained significant attention in recent years due to their impressive performance in various natural language processing (NLP) tasks Zhang et al. (2022); Penedo et al. (2023); Touvron et al. (2023); Zhou et al. (2023); Liu et al. (2019). However, their training process often relies on computationally intensive procedures that involve massive datasets and compute requirements which hinders training large scale LMs on noisy real-world or domain specific datasets. What's worse is that several of these datasets are uncurated and may contain harmful content which the LM model can potentially pick up during the training process Deshpande et al. (2023); Schramowski et al. (2022); Kuchnik et al. (2023).

Text quality evaluation plays a crucial role in assessing the suitability and reliability of textual data for training LMs. Previous research has explored various approaches for text quality assessment, primarily focusing on human annotation and subjective judgments. For instance, Clark et al. (2021) introduce a crowdsourcing-based method for ranking text quality, where human evaluators provide subjective ratings. While such approaches provide valuable insights, they suffer from scalability limitations and subjectivity biases. To overcome these limitations, more recent works have explored the use of automated approaches to quality evaluation such as making use of ChatGPT or GPT-4 to evaluate the quality of the text, where text is designated to be high quality if ChatGPT/GPT-4 deems it to be similar to human text Gilardi et al. (2023); Liu et al. (2023). However, these methods are model dependent and requires training massive LLM models, which defeats the purpose of efficient LM training.

We address this issue by proposing a novel method for numerically evaluating text quality in large unlabelled NLP datasets, with the aim of improving LM training performance and efficiency. We also ensure that our text quality metric is model agnostic, helping us avoid having to recompute these quality metrics for each model. By leveraging this numerical text quality score, we demonstrate how it can be used to prune the original dataset, enabling the training of LMs using only a fraction of the data. Our approach aims to identify and eliminate low-quality text instances, thereby streamlining the training process and mitigating the burden of handling large-scale datasets. We also remove potentially harmful content from the data by ensuring that harmful content is rated poorly by our text quality score which can then be pruned. We observe an absolute improvement of 0.9% averaged over 14 downstream evaluation tasks for multiple LM models while using 40% lesser data and training 42% faster when training on the OpenWebText dataset Gokaslan et al. (2019) and a 0.8% absolute improvement averaged over 3 models and 14 downstream tasks for the Wikipedia dataset Tunstall et al. when using 20% lesser data and training time .

The key contribution of this paper lies in establishing a framework that quantitatively evaluates text quality in a model agnostic manner and subsequently guides the pruning of

NLP datasets for LM training. By leveraging this quality score metric, we enable a more efficient allocation of computational resources and reduce the data requirements for training LMs. This approach not only expedites the training process but also enhances the overall effectiveness of the models. To the best of our knowledge, there doesn't exist an objective way to evaluate the quality of large scale textual datasets and we hope this work will pave the way for further work in this space.

## 2 Methodology

### 2.1 Computing Text Quality

The notion of text "quality" is a fairly ambiguous one. Presently, no concrete and objective method exists for quantitatively evaluating data quality. In this section, we combine commonly used heuristics from literature to formulate a comprehensive definition for text quality. We presently demonstrate the effectiveness of our approach on only English text but the filters and method can be easily extended to other languages. Our proposed method has 2 steps:

- **Weight calculation**: In this step we use 14 heuristic based filters covering a wide range of linguistic characteristics like text complexity (measured using parse tree depth and structure), word repetition ratio, syntax of the text (based on presence and relation between objects, nouns and determiners), text length etc. The full list of heuristic filters are listed in Table 2 that identify text that follow attributes of a well formed sentence. We apply each filter individually on a dataset to obtain a data subset corresponding to each filter with text instances qualifying that specific filter. These subsets are used as evaluation datasets for a pre-trained LM to calculate validation perplexity ($PPL$) for filtered subsets and the original unfiltered dataset. We implement our filters using spacy Honnibal et al. (2020) and for validation perplexity calculation we use HuggingFace based pre-trained language modelWolf et al. (2020). We then use following formulation to calculate weight for each heuristic:

$$w_i = max(0, \frac{PPL_{all} - PPL_i}{PPL_{all}}) \tag{1}$$

Here $w_i$ is weight for the $i^{th}$ filter where $i = 1, 2, ...14$, $PPL_i$ is the perplexity for the subset created after applying filter $i$ and $PPL_{all}$ is the perplexity of the unfiltered dataset. We lower bound the weights to 0 for filters where $PPL$ goes up to avoid negative weights. The chosen 14 filters were selected from a diverse set of over a 50+ filters based on consistent perplexity improvements, leading to them consistently being assigned a higher weight. The final weights assigned to each of the filters are presented in Figure 2. The simplicity of the chosen filters make it extremely fast to compute these quality scores while increasing their generalization abilities across datasets. For example, it takes 26.41s to compute the scores for 10k lines of text on a single CPU core, and the computation could be easily parallelized across multiple cores while observing linear speedup in throughput with number of cores.

- **Quality scoring**: In this step each document in the dataset is split into lines based on common sentence end markers like period or HTML end tags and for each line all

the heuristic filters are applied that results in an indicator matrix $I$ where $I_i(line) = 1$ indicates that $line$ satisfies the $i^{th}$ filter criteria. Then we use the weights calculated in the above step to get quality score per line. This can be formulated as:

$$score_{line} = \frac{\sum_{i=1}^{F} w_i I_i(line)}{\sum_{i=1}^{F} w_i} \qquad (2)$$

Here $score_{line}$ is the quality score assigned to $line$, $w_i$ is the weight for filter $i$, $F$ is the number of filters we use and $I_i$ is the indicator function for filter $i$.

We then aggregate the scores for each line in the document to obtain document level score by taking a weighted average of scores of each line in the document, where the line weights are proportional to the token length of the line. Following is the formulation used for the doc score:

$$score_{doc} = \frac{\sum_{line=1}^{n} tc_{line} score_{line}}{\sum_{line=1}^{n} tc_{line}} \qquad (3)$$

Here $score_{doc}$ is the aggregated quality score for the doc, $tc_{line}$ is the token count for the line, $score_{line}$ is the score for the line calculated as per equation 2 and $n$ is the total count of lines in the doc.

Our method is completely model agnostic and relies solely on the underlying data and hence can be generalized to the training of any downstream LM model.

## 2.2 Quality guided data pruning

With the computed text quality scores, we prune the dataset by selecting the desired fraction of the dataset by retaining highest quality samples. The threshold can be determined based on the specific requirements of the LM training task and the available computational resources. Instances with text quality scores below the threshold are considered low-quality and are removed from the dataset. The remaining high-quality instances form the pruned dataset for subsequent LM training. By training the LM on the pruned dataset, we demonstrate that the model can achieve comparable or even improved performance with significantly fewer training instances, leading to improved LM training. In this work we use percentile based pruning, where we select data subset with quality score in top 20%, 40%, 60% and 80% and compare its performance to the models trained on the unpruned datasets as baseline.

## 3 Experimental Details

### 3.1 Datasets

We experiment with a english only versions of following datasets for our study:

- **Wikipedia** Tunstall et al. : This dataset is built from the wikipedia dump where each sample contains whole wikipedia article. This dataset contains 4.67 billion tokens before pruning or splitting into train and validation sets.

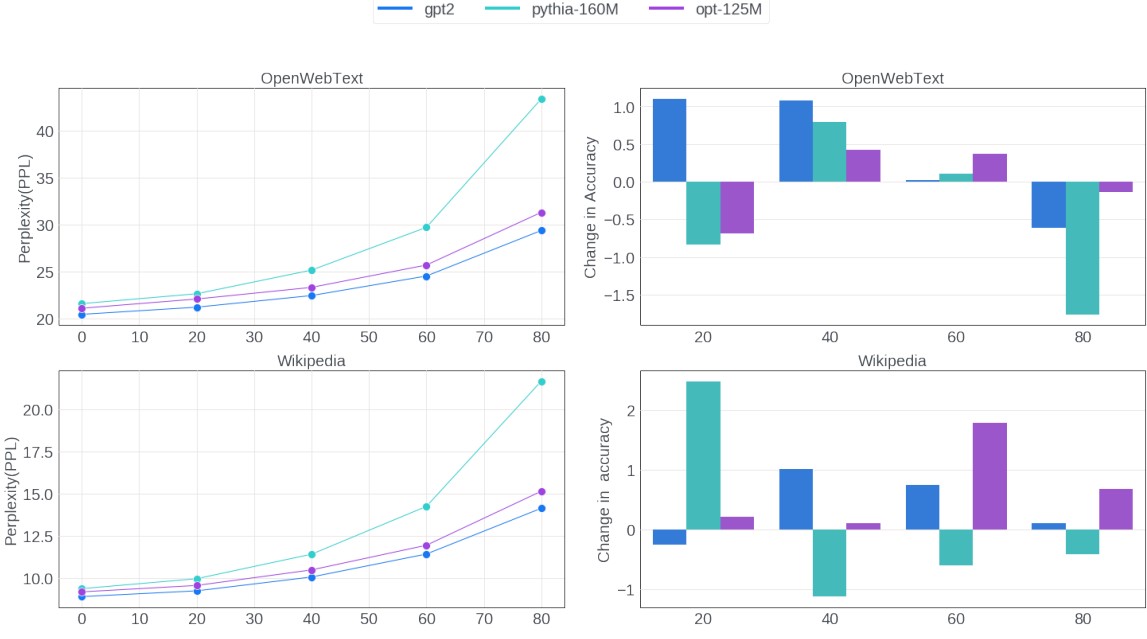

Figure 1: Change in accuracy for pruned datasets compared to no pruning for OpenWeb-Text and Wikipedia data

- **OpenWebtext** Gokaslan et al. (2019): This dataset is the open source version of the WebText dataset used for GPT-2 training. To build this dataset Reddit post urls from Reddit submission dataset with html content were used. The base version of the dataset contains 9.03 billion tokens.

| Text | Quality score |
|------|------|
| [Accessories](/directory/Shopping/Accessories/49511) | 0.12 |
| Champions of Bundaberg Touch competition](¡url¿ touch-competition/) | 0.26 |
| [Microsoft 365: Get OneDrive for Business Usage Report using PowerShell](¡url¿ "Microsoft 365: Get OneDrive for Business Usage Report using PowerShell") | 0.45 |
| We have no tolerance for comments containing violence, racism, profanity, vulgarity, doxing, or discourteous behavior. If a comment is spam, instead of replying to it please click the icon below and to the right of that comment. Thank you for partnering with us to maintain fruitful conversation. | 0.68 |
| You're one among a lucky few. You found your love in a guy of another culture! I know a distant relative of mine who married a black woman from a developed nation. They loved each other, married and settled in her country. At one point, he was asked to leave her by his family and marry an Indian instead, but he said he would never be able to leave her for another. How amazing! Now they're old, retired and live in India, but still love each other nevertheless. | 0.89 |

Table 1: Samples of lines with assigned quality scores.

## 3.2 Models

To ensure the consistency and generalizability of our study, we experiment with a diverse set of popular models including GPT2 Radford et al. (2019), GPT-Neo-125M Black et al. (2022), Pythia-160M Biderman et al. (2023) and OPT-125M Zhang et al. (2022). All the models are trained from scratch with 15 epochs and batch size of 128, we use HuggingFace trainer to train our models.

## 3.3 Evaluation

We follow the evaluation setup consistent with OPTZhang et al. (2022). We calculate validation perplexity for each of the dataset where validation set is 20% of the whole dataset sampled before pruning and is removed from the training data used for pruning. We also evaluate 0-shot accuracy of all trained models on 14 downstream NLP tasks. These 14 NLP tasks include Arc Challenge and ARC EasyClark et al. (2018), HellaSwagZellers et al. (2019), OpenBookQAMihaylov et al. (2018), PIQABisk et al. (2019), StoryClozeSchwartz et al. (2017), WinogradLevesque et al. (2012), WinograndeSakaguchi et al. (2019) and tasks from SuperGLUEWang et al. (2020). We use lm-evalaution-harnessGao et al. (2021) to downstream task based evaluation.

## 4 Results and Analysis

We compute the text quality score for the OpenWebText and Wikipedia datasets. Table 1 shows some samples texts from these datasets and the text quality scores they get assigned based on our method. As can clearly be seen, the higher quality sentences in terms of content, grammatical and linguistic quality do seem to consistently be rated as higher quality by our approach.

Next we analyze the results obtained from our pruning experiments using data quality as a measure to eliminate lower quality samples. Figure 1 presents the average change in accuracy (%) using the model trained on the unpruned datasets as the baseline. The accuracy is averaged over the 14 downstream tasks as explained in the previous section. Variations in individual task accuracies are presented in the Appendix. As can be seen, for most models, the performance seems to improve with lower pruning levels up to a threshold and then declines sharply. For OpenWebText, most models achieve peak performance at around 40% pruning level while the same can be seen for Wikipedia data at around 20% pruning level. This points to the presence of a subset of low quality data in these datasets, which can be removed from model training without affecting downstream model performance while significantly improving data efficiency and the time needed to train these models. Note that the trends as observed in downstream model performance are consistent yet a little noisy, as has often been observed in prior literature Zhang et al. (2022); Wang et al. (2020).

We further analyze the variation in perplexity over the validation set for GPT2 Radford et al. (2019), Pythia-160M Biderman et al. (2023) and OPT-125M Zhang et al. (2022) trained over different pruning levels for both OpenWebText and Wikipedia reveal a consistent trend of perplexity of the trained LM models increasing with more data being pruned as can be seen in Figure 1. The increase in perplexity is fairly gradual to a certain level (20%

for Wikipedia and 40% for OpenWebText) and then increases significantly faster beyond that pruning level. The sudden increase in perplexity beyond a threshold points to the fact that the data being pruned after that threshold is potentially high quality data.

The contributions of our work extend beyond the immediate scope of LM training. The introduced text quality evaluation framework provides a foundation for further advancements in the field, enabling researchers to objectively assess the quality of large-scale textual datasets. This paves the way for future research on improving data curation, dataset selection, and the development of automated methods for text quality assessment.

## Limitations

While our research provides promising results and demonstrates the effectiveness of text quality evaluation and dataset pruning for improving the training efficiency of Language Models (LMs), there are several limitations that should be considered. These limitations highlight the potential areas for further investigation and exploration in future research.

### 4.1 Generalizability to Larger Models

One limitation of our work is that we primarily focus on LM models with a relatively smaller number of parameters. The effectiveness of our approach needs to be further tested and validated on much larger models, such as models with hundreds of billions of parameters like Falcon40B Almazrouei et al. (2023), LLaMa Touvron et al. (2023), OPT-175B Zhang et al. (2022) among others. Larger models often exhibit different training dynamics and may require different considerations when it comes to dataset pruning. Therefore, future research should investigate the scalability and applicability of our methodology to such larger models.

### 4.2 Scalability to Larger Datasets

Another limitation is the scale of the datasets used in our experiments. While we have conducted experiments on large-scale datasets, future research should explore the effectiveness of our approach on even larger datasets, involving billions of samples like the Pile dataset Gao et al. (2020). Training LLM models on such massive datasets poses unique challenges in terms of computational resources, data storage, and training time. Evaluating the scalability and practicality of our approach on such datasets will provide a more comprehensive understanding of its potential benefits and limitations.

### 4.3 Evaluation Metrics and Robustness

We have primarily evaluated the effectiveness of our approach based on standard evaluation metrics such as perplexity on the validation set and accuracy on 14 downstream evaluation tasks. However, the evaluation of LM models goes beyond these metrics, and future research should explore additional evaluation criteria such as robustness, fairness, and interpretability. Understanding the impact of dataset pruning on these aspects will provide a more comprehensive assessment of our approach's efficacy.

## Ethics Statement

While our work addresses the issue of harmful content in datasets through the application of text quality evaluation, ethical considerations surrounding bias, fairness, and inclusivity in LM training remain significant challenges. Further research is needed to develop methodologies that effectively address these ethical concerns and ensure the responsible deployment of LM models in real-world applications.

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

# 5 Appendix

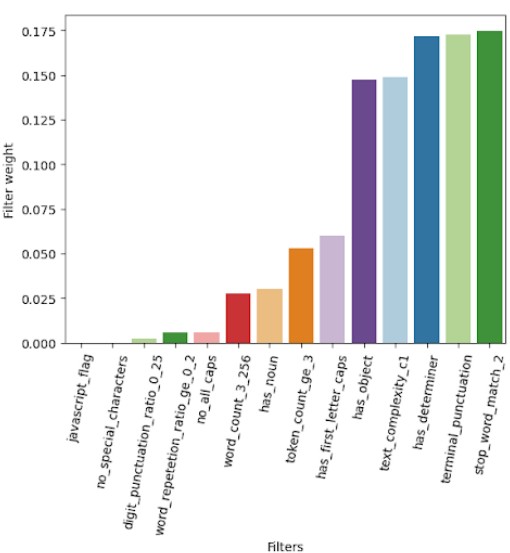

Figure 2: Assigned weights for all the filters.

| filter name | Heuristic | Description |
|---|---|---|
| has_first_letter_caps | First character capitalized | Check if first character of each line is capitalized. |
| no_all_caps | All characters capitalised | Check if all the characters in the line are capitalized |
| word_repetetion_ratio_ge_0_2 | Word repetition ratio | Check if ratio of repetition for word in line is ¿ 0.2 |
| digit_punctuation_ratio_0_25 | Digit/punctuation to word ratio | Identify lines with ratio of digits/punctuation to words in a line is ¿ 0.25. |
| no_special_characters | Has { character | Flower brackets are usually common in code as we are curating for text only content this filter identifies text that might contain code. |
| terminal_punctuation | Has terminal punctuation | Check if the lines end with one of these puntuation marks - '.', '!', '?', '"'. |
| stop_word_match_2 | Has 2 stop words | Check if the line contains at least 2 stop words among 'the', 'be', 'to', 'of', 'and', 'that', 'have', 'with'. |
| javascript_flag | Contains special phrases | Check if text contains phrases 'javascript' or 'lorem ipsum' to identify docs with code. |
| token_count_ge_3 | Token count | Check if the token count is ¿ 3 |
| word_count_3_256 | Word count range | Check if line word count is ¿ 3 and ¡ 256. |
| has_object | Has object | check if there is object identified by parser. |
| has_noun | Has noun | Check if there is at least one noun in the line. |
| has_determiner | Has determiner | Check if the line contains determiner based on results from text parser. |
| text_complexity_c1 | Text complexity | For this we use setup similar to CAT filterRadenovic et al. (2023), where lines with atleast one edge from object are flagged as positive. |

Table 2: Set of heuristics used for quality score calculation.

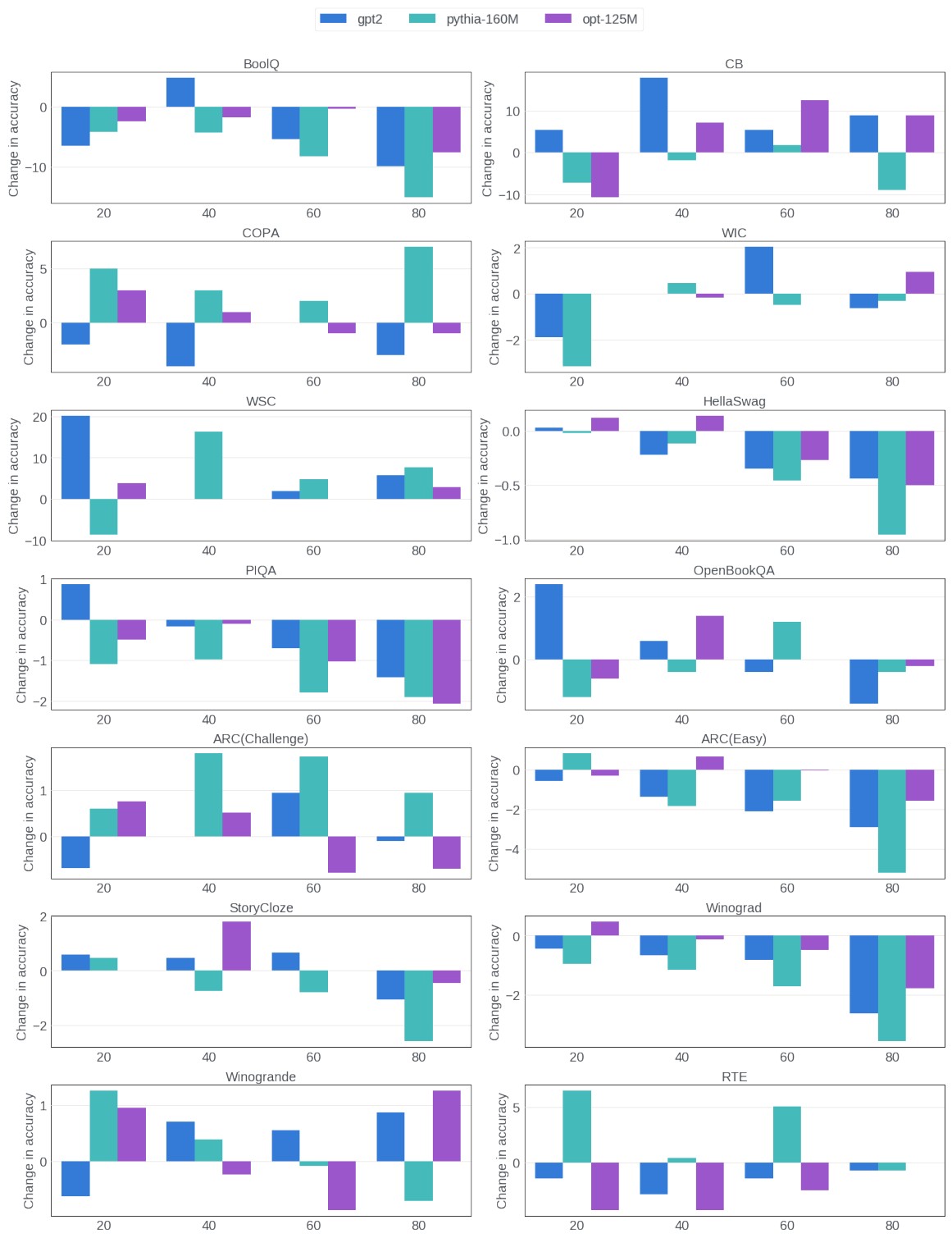

Figure 3: Change in accuracy of models trained on pruned data compared to unpruned data for all the 14 tasks on OpenWebText

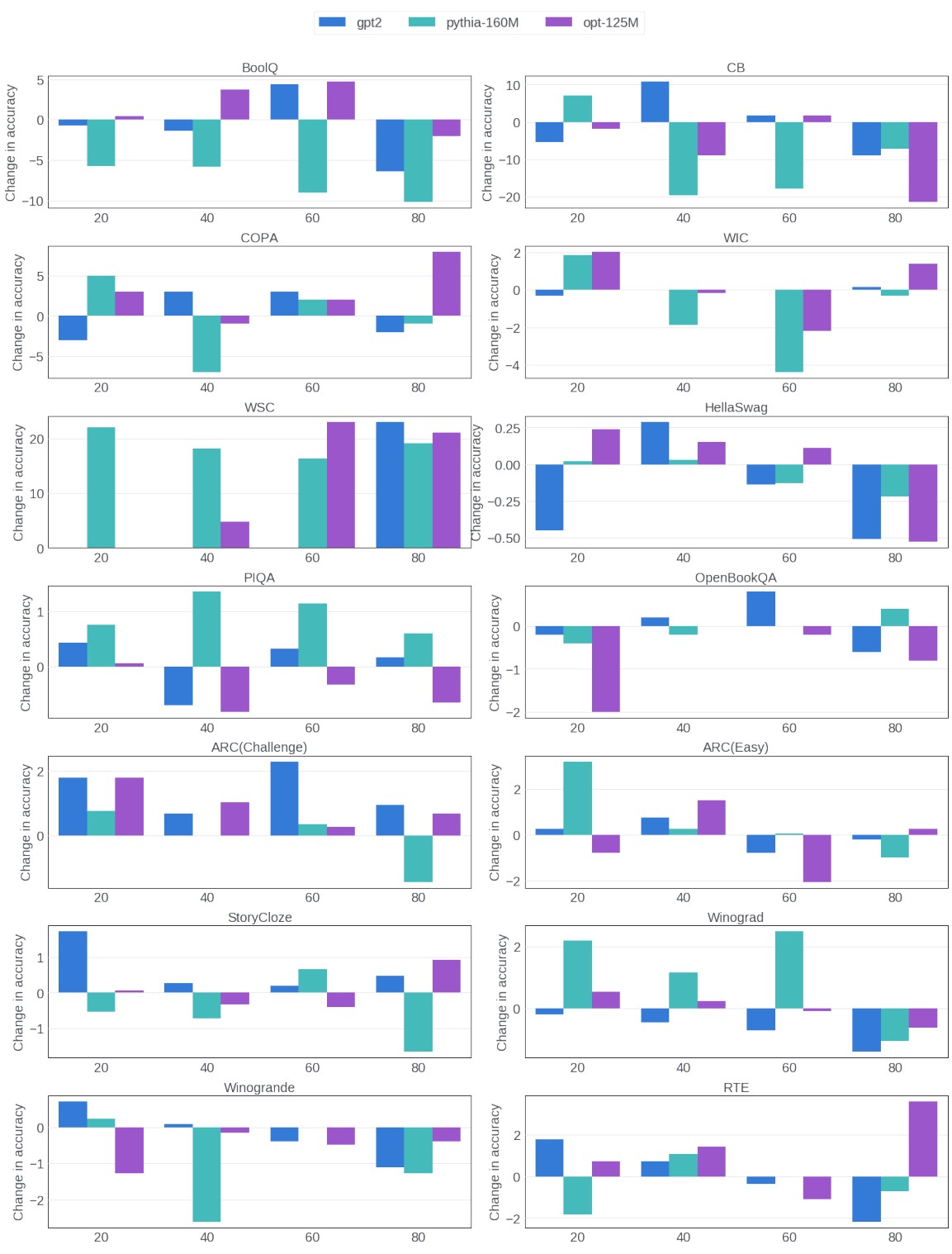

Figure 4: Change in accuracy of models trained on pruned data compared to unpruned data for all the 14 tasks on Wikipedia

