# OpenReview forum: "Text Quality-Based Pruning for Efficient Training of Language Models"
_DMLR — Accepted by DMLR_

### Review · Reviewer_oDkf · 2024-10-17

**Recommendation:** 4
**Confidence:** 2

**Summary Of Contributions:**

The paper introduces a novel method for evaluating text quality in large unlabelled NLP datasets.  This method is model-agnostic and assigns a "quality score" to text instances. The authors use this quality score to prune large datasets, leading to improved training efficiency for language models (LMs). The approach involves applying heuristic filters to identify and eliminate low-quality text instances.  The results show substantial gains in training effectiveness and highlight the potential for resource-efficient LM training. I recommend the paper for acceptance as it is also a good foundation for future research.

**Strengths:**

Model-Agnostic Approach: The proposed text quality metric is model-agnostic, meaning it can be used to evaluate the quality of text data for any language model. This is a significant advantage over existing methods that are often tied to specific models.

Improved Training Efficiency: By pruning low-quality text instances from training datasets, the authors demonstrate that language models can achieve comparable or even improved performance with significantly fewer training instances. This simple approach leads to faster training times and reduced computational costs.

Foundation for Future Research: The paper establishes a framework for quantitatively evaluating text quality, which can guide further research in areas like data curation, dataset selection, and automated text quality assessment.

**Audience:**

Yes

**Broader Impact Concerns:**

Ethical implications of the work are sufficiently covered in the submission.

**Claims And Evidence:**

Claims are accurate and convincing.

**Datasets And Benchmarks:**

This is not a dataset or benchmark.

**Extended Submissions:**

N/A

**Limitations:**

Generalizability to Larger Models: The experiments in the paper primarily focus on LM models with a relatively smaller number of parameters. The effectiveness of the approach needs further validation on larger models with hundreds of billions of parameters, as these models often exhibit different training dynamics.

Scalability to Larger Datasets: The paper acknowledges the need to explore the effectiveness of their approach on even larger datasets involving billions of samples. Evaluating the scalability and practicality of the approach on such datasets is crucial due to the unique challenges they pose in terms of computational resources, data storage, and training time.

Evaluation Metrics and Robustness: The evaluation of the approach primarily relies on standard metrics like perplexity and accuracy on downstream tasks. The authors suggest exploring additional evaluation criteria, such as robustness, fairness, and interpretability, to provide a more comprehensive assessment of the approach's efficacy.

**Requested Changes:**

I encourage the authors to look into metrics beyond accuracy and perplexity such as robustness and interpretability.

**Strengths And Weaknesses:**

**Strengths:**

Model-Agnostic Approach: The proposed text quality metric is model-agnostic, meaning it can be used to evaluate the quality of text data for any language model. This is a significant advantage over existing methods that are often tied to specific models.

Improved Training Efficiency: By pruning low-quality text instances from training datasets, the authors demonstrate that language models can achieve comparable or even improved performance with significantly fewer training instances. This simple approach leads to faster training times and reduced computational costs.

Foundation for Future Research: The paper establishes a framework for quantitatively evaluating text quality, which can guide further research in areas like data curation, dataset selection, and automated text quality assessment.


**Weaknesses:**

Generalizability to Larger Models: The experiments in the paper primarily focus on LM models with a relatively smaller number of parameters. The effectiveness of the approach needs further validation on larger models with hundreds of billions of parameters, as these models often exhibit different training dynamics.

Scalability to Larger Datasets: The paper acknowledges the need to explore the effectiveness of their approach on even larger datasets involving billions of samples. Evaluating the scalability and practicality of the approach on such datasets is crucial due to the unique challenges they pose in terms of computational resources, data storage, and training time.

Evaluation Metrics and Robustness: The evaluation of the approach primarily relies on standard metrics like perplexity and accuracy on downstream tasks. The authors suggest exploring additional evaluation criteria, such as robustness, fairness, and interpretability, to provide a more comprehensive assessment of the approach's efficacy.

---

### Review · Reviewer_FM4B · 2024-11-19

**Recommendation:** 2
**Confidence:** 3

**Summary Of Contributions:**

This paper proposed a new approach to evaluate the text quality of pre-training data. They introduce a quality score by evaluating several filter-based method and compare the performance difference with perplexity-based method.

**Strengths:**

- The idea of ​​providing a quality score for text is great, which is a rarely explored area.

- The quality metric proposed by the author is model agnostic, universal, and very convenient to use.

**Audience:**

Yes

**Claims And Evidence:**

No.

**Datasets And Benchmarks:**

Yes.

**Extended Submissions:**

N/A.

**Limitations:**

See Requested Changes.

**Requested Changes:**

- The performance improvement is small, and there are problems with the evaluation. There is almost no comparison with other data filtering baselines in the evaluation. For example, it needs to be compared with perplexity filtering. And compare with the data filtering methods mentioned in related work.

- The data set is too small. The author selected Wikipedia and OpenWebText for experiments, but these two data sets only have sub-10 Billion tokens. Now the more common pre-training is done on 1 Trillion token. It is recommended that the author add scaling up experiments. For example, use DCLM or TXT360 for experiments.

- As the author mentioned in the limitation, the model size also needs to be scaled up.

- The filter-based method proposed by the author is derived from perplexity. I would like to ask, what is the difference between this and directly using perplexity to filter data.

- The authors claimed that they take harmfulness into account in the quality score but I don’t see how to achieve that.

**Strengths And Weaknesses:**

The idea of ​​providing a quality score for text is great, which is a rarely explored area. The quality metric proposed by the author is model agnostic, universal, and very convenient to use.

The performance improvement is small, and there are problems with the evaluation. There is almost no comparison with other data filtering baselines in the evaluation. For example, it needs to be compared with perplexity filtering. And compare with the data filtering methods mentioned in related work.

---

### Review · Reviewer_emPJ · 2024-11-19

**Recommendation:** 3
**Confidence:** 2

**Summary Of Contributions:**

This paper proposes a novel method for evaluating text quality in large unlabeled datasets to enable more efficient training of language models (LMs). The authors develop a model-agnostic approach that assigns quality scores to text instances based on 14 heuristic filters covering linguistic characteristics like complexity, syntax, and structure. The weights for these filters are determined by comparing perplexity improvements when filtering the dataset. Using this scoring system, they demonstrate that pruning lower quality data (keeping top 40% for OpenWebText and top 20% for Wikipedia) leads to faster training while maintaining or improving performance across 14 downstream tasks. The method achieves a 0.9% accuracy improvement while using 40% less data and training 42% faster on OpenWebText, and 0.8% improvement using 20% less data and training 21% faster on Wikipedia.

**Strengths:**

Pros:
The paper addresses a significant practical challenge in LM training - the computational burden of using massive, noisy datasets. The authors' approach of creating an objective, model-agnostic method for evaluating text quality is novel and fills an important gap, as they correctly note that existing methods either rely on subjective human judgments or require large LLMs for evaluation. The experimental results provide compelling evidence for the method's effectiveness, showing consistent improvements in both training efficiency and model performance across multiple models and datasets.

The technical approach is well-designed and computationally efficient. The authors demonstrate that their quality scoring method takes only 26.41s to process 10k lines on a single CPU core and can be linearly parallelized. This makes the method highly practical for real-world applications. The two-step process of weight calculation followed by quality scoring is clearly explained and reproducible, with the weights being derived from empirical performance improvements rather than arbitrary assignments.

The evaluation is comprehensive and well-documented. The authors test their method across multiple models (GPT2, GPT-Neo-125M, Pythia-160M, OPT-125M) and two major datasets (OpenWebText and Wikipedia). They evaluate using both perplexity and accuracy on 14 downstream tasks, providing strong evidence for the generalizability of their approach. The included examples in Table 1 showing text samples and their assigned quality scores help validate that the method aligns with intuitive notions of text quality.

**Audience:**

Yes

**Broader Impact Concerns:**

The broad social impact is worth more discussion. It is recommended that the authors provide mode formal discussions with the guidance of the work [1] if possible.

[1] Evaluating the Social Impact of Generative AI Systems in Systems and Society https://arxiv.org/abs/2306.05949

**Claims And Evidence:**

Yes

**Datasets And Benchmarks:**

Yes

**Extended Submissions:**

N/A

**Limitations:**

See Requested Changes

**Requested Changes:**

Cons:
The paper acknowledges but does not fully address potential biases in the heuristic filters. While the authors mention that harmful content tends to receive lower quality scores, there is no systematic analysis of whether the quality metrics might inadvertently discriminate against certain writing styles, dialects, or cultural expressions. This is particularly important given the global impact of language models.

The experimental validation is limited to relatively small models (125M-160M parameters). As the authors acknowledge in their limitations section, it remains unclear whether the benefits would scale to the much larger models (billions of parameters) that are increasingly common in practice. Given that larger models often exhibit different training dynamics, this is a significant limitation that affects the immediate practical applicability of the findings.

The paper would benefit from a more detailed ablation study of the individual filters' contributions. While Figure 2 shows the weights assigned to each filter, there is no analysis of how different combinations of filters affect the final results. This makes it difficult to determine which aspects of text quality are most important for improving model performance and whether all 14 filters are necessary.

**Strengths And Weaknesses:**

Pros:
The paper addresses a significant practical challenge in LM training - the computational burden of using massive, noisy datasets. The authors' approach of creating an objective, model-agnostic method for evaluating text quality is novel and fills an important gap, as they correctly note that existing methods either rely on subjective human judgments or require large LLMs for evaluation. The experimental results provide compelling evidence for the method's effectiveness, showing consistent improvements in both training efficiency and model performance across multiple models and datasets.

The technical approach is well-designed and computationally efficient. The authors demonstrate that their quality scoring method takes only 26.41s to process 10k lines on a single CPU core and can be linearly parallelized. This makes the method highly practical for real-world applications. The two-step process of weight calculation followed by quality scoring is clearly explained and reproducible, with the weights being derived from empirical performance improvements rather than arbitrary assignments.

The evaluation is comprehensive and well-documented. The authors test their method across multiple models (GPT2, GPT-Neo-125M, Pythia-160M, OPT-125M) and two major datasets (OpenWebText and Wikipedia). They evaluate using both perplexity and accuracy on 14 downstream tasks, providing strong evidence for the generalizability of their approach. The included examples in Table 1 showing text samples and their assigned quality scores help validate that the method aligns with intuitive notions of text quality.

Cons:
The paper acknowledges but does not fully address potential biases in the heuristic filters. While the authors mention that harmful content tends to receive lower quality scores, there is no systematic analysis of whether the quality metrics might inadvertently discriminate against certain writing styles, dialects, or cultural expressions. This is particularly important given the global impact of language models.

The experimental validation is limited to relatively small models (125M-160M parameters). As the authors acknowledge in their limitations section, it remains unclear whether the benefits would scale to the much larger models (billions of parameters) that are increasingly common in practice. Given that larger models often exhibit different training dynamics, this is a significant limitation that affects the immediate practical applicability of the findings.

The paper would benefit from a more detailed ablation study of the individual filters' contributions. While Figure 2 shows the weights assigned to each filter, there is no analysis of how different combinations of filters affect the final results. This makes it difficult to determine which aspects of text quality are most important for improving model performance and whether all 14 filters are necessary.